# Assessing the Nutrient Composition of a Carnivore Diet: A Case Study Model

**DOI:** 10.3390/nu17010140

**Published:** 2024-12-31

**Authors:** Sylvia Goedeke, Tamzyn Murphy, Amy Rush, Caryn Zinn

**Affiliations:** 1Human Potential Centre, Faculty of Health and Environmental Sciences, Auckland University of Technology, Auckland 0632, New Zealand; sylvia.goedeke@aut.ac.nz; 2Nutrition Network, Cape Town, South Africa; 3Type 1 Diabetes Family Centre, Stirling 6021, WA, Australia; amy@type1familycentre.org.au

**Keywords:** animal-based diet, elimination diet, carnivore, nutrient analysis, dietetics

## Abstract

Background/Objectives: The rise in chronic metabolic diseases has led to the exploration of alternative diets. The carnivore diet, consisting exclusively of animal products, has gained attention, anecdotally, for imparting benefit for inflammatory conditions beyond that possible by other restrictive dietary approaches. The aim was to assess the micronutrient adequacy of four versions of the carnivore diet against national nutrient reference values (NRVs). Methods: This study assessed the nutrient adequacy of the carnivore diet against national NRVs from the Australian National Health and Medical Research Council (NHMRC) and New Zealand Ministry of Health. Four meal plans for hypothetical average Australian adults were developed and analysed using Foodworks.online (Version 1, Xyris Pty Ltd., Brisbane, Australia, 2024), dietary software. Two female and two male plans were included; one set including dairy products and the other set including offal. Results: The carnivore diet met several NRV thresholds for nutrients such as riboflavin, niacin, phosphorus, zinc, Vitamin B6, Vitamin B12, selenium, and Vitamin A, and exceeded the sodium threshold. However, it fell short in thiamin, magnesium, calcium, and Vitamin C, and in iron, folate, iodine and potassium in some cases. Fibre intake was significantly below recommended levels. Conclusion: The carnivore diet may offer benefits for managing certain chronic conditions. Whether the metabolic contexts from consuming such a diet facilitates a lower requirement of certain nutrients, or whether it poses risks of micronutrient inadequacies remains to be determined. Tailored nutritional guidance and supplementation strategies are recommended to ensure careful consideration of micronutrient intake to prevent deficiencies.

## 1. Introduction

With the global modern world rise in chronic metabolic disease, traditional dietary management paradigms are persistently being challenged. Over the last decade, alternative nutritional strategies, including some highly restrictive dietary approaches, have gained traction for their therapeutic application in the context of inflammatory conditions. Examples include the auto-immune protocol (AIP) diet for auto-immune conditions [1,2], a low FODMAPs diet for irritable bowel syndrome [3], and carbohydrate reduction including the more extreme ketogenic diet for type 2 diabetes [4] and inflammatory neurological conditions [5]. While previously being considered contemporary clinical practice, carbohydrate reduction is now recognised as a valid and effective therapeutic option for the management of type 2 diabetes and is endorsed within consensus guidelines by several organisations including Diabetes UK [6,7], the American Diabetes Association [8], European Association for the Study of Diabetes [9], Diabetes Canada [10], Diabetes Australia [11], and the Australian Diabetes Society [12].

Recently, there has been growing interest in another contemporary dietary strategy, a carnivore diet [13]. This diet is based exclusively on consuming animal products, and serves on the basis that eliminating all plant antigens and xenobiotic secondary metabolites may have immunological and inflammatory benefits beyond that achieved by a standard low-carbohydrate or ketogenic diet [14]. While there is a substantial body of literature to support the benefits of carbohydrate reduction on metabolic health [4], claims surrounding additional anti-inflammatory benefits following the elimination of plant-based foods have not been founded by peer-reviewed literature. It is also a common view that high consumption of animal products may have negative health effects due to changes in the microbiota [15], higher cardiovascular disease risk profile [16], and other chronic health issues [15], however, the scientific basis for many of these potential negative effects is increasingly being questioned. To date, research examining the contemporary carnivore diet is extremely limited.

Despite in many ways being contrary to mainstream dietary advice, the carnivore dietary strategy has become increasingly popular in recent years. Lennerz, Mey [13] presented a survey of a social media group consisting of over 2000 adults who self-identified as following a Carnivore Diet [13]. Participants indicated that following a diet pattern with minimal to no plant matter led to resolution or improvement in a range of chronic conditions, including overweight/obesity, diabetes/insulin resistance, gastrointestinal conditions, and dermatologic conditions after following a completely animal-based diet [13]. Participants with diabetes reported marked reductions in exogenous insulin and oral diabetic medications, as well as relatively large reductions in BMI and glycated haemoglobin. Although the survey was vulnerable to a positive user bias, participants reported very few adverse effects and instead indicated a high level of satisfaction and overall health benefits from consuming a carnivore diet.

It is well-accepted that dietary patterns which exclude food groups must take nutrient adequacy into consideration. Both the vegetarian and vegan diets are styles of eating that exclude animal products to varying degrees, which can result in nutrient inadequacies. For example, Vitamin B_12_, riboflavin (B_2_), niacin (B_3_), long-chain omega-3-fatty acids, iron, calcium, zinc, iodine, selenium, and fat-soluble vitamins pre-formed Vitamin A and D intakes are typically lower in vegan diets [17,18]. Dietary inadequacies are commonly addressed with supplementation to evade long-term nutritional issues [19].

Carbohydrate-reduction has faced similar criticism as being a dietary approach that also restricts quantities of certain foods. In 2018, our group presented a nutritional analysis of a carbohydrate-reduction dietary approach in hypothetical case studies [20]. The meal plans were shown to be replete in all micronutrients according to the national Australian/New Zealand nutrient reference values (NRV) [21], apart from iron in the female meal plans, which achieved 86–98% of the threshold. Understandably, the carnivore diet is eligible to receive similar criticisms as other ketogenic/carbohydrate-reduced dietary patterns for excluding certain food groups. A recent perspective suggests that all essential nutrients can be obtained from a carnivore diet [14]; however, there is no published literature examining actual nutrient intakes of individuals following the diet. Therefore, the aim of this study was to assess the micronutrient adequacy of four versions of the carnivore diet against national NRV thresholds, as set by the Australian National Health and Medical Research Council (NHMRC) and New Zealand Ministry of Health (MOH) [21].

## 2. Materials and Methods

In this descriptive study, we designed a total of four carnivore meal plans, two for each of two theoretical case studies representing the average Australian male and female as closely as possible. Using the body mass index (BMI) equation, BMI = (weight-kg)/(height-m)^2^, we inputted national median heights recorded by the Australian Bureau of Statistics [22] for male and female adults (aged 25–54 years) and used the midpoint for a healthy range BMI of 22.5 to calculate body weight (See Table 1). To estimate total energy expenditure, we used the Schofield equation [23], where weight and height variables and an activity factor of 1.6 (light level) were inputted. The adult age range category of 19–50 years was selected from the Australian NHMRC and New Zealand MOH set of categories [21]. The only NRV that varies within this age category is magnesium, which is slightly higher, for both males and females in the 35–50 years age subcategory than in the 19–35 year age subcategory; this was accounted for in the analysis. Total energy intake would likely change within the age groups and was accounted for in the analysis by ensuring that energy intake was met within 95% of requirements for both a 25-year-old and a 45-year-old male and female, respectively. We assumed weight stability and matched the energy intake with the calculated energy expenditure.

There is no universal definition for the Carnivore diet or specific parameters for macronutrient proportions. Broadly, the diets can typically range between 60–80% energy from fat and 20–40% energy from protein, with the remaining carbohydrate intake usually being below 5% of total energy. We approximated definitions of macronutrient proportions in the meal plans using a combination of anecdotal evidence, practice/experience from dietitians familiar with the approach, and practical food considerations. The threshold protein intake was selected as 25–30% of energy intake, which is above the acceptable macronutrient distribution range (AMDR) as per the Australian NHMRC and New Zealand Ministry of Health guidelines (25% of total energy intake). Since the plans only consist of animal products, a higher protein intake is generally expected with this dietary approach. Fat was set at the remaining calories at 70–75% of energy intake. Carbohydrate intake was set at <5% of energy intake, givn that the diet typically includes minimal carbohydrate-containing foods. Estimated energy requirement was calculated by applying demographic characteristics provided in Table 1 to the Schofield equation [23] with a physical activity level of 1.6. Energy and macronutrient parameters for the theoretical case studies are shown in Table 2.

The NRVs are a set of recommendations for nutritional intake based on currently available scientific knowledge. NRVs include recommended dietary intakes (RDIs), a term used to represent the average daily micronutrient level sufficient to meet the requirements of almost all (97–98%) healthy individuals of a certain gender and life stage, and adequate intakes (AIs), where the nutrient intake level is based on observed or experimentally determined nutrient estimates of apparently healthy people and are assumed to be adequate. Where the RDI value was not available, the AI value was used.

We created two different meal plans each for the female and male case studies for the purpose of demonstrating variety in food options based on different versions of the diet. We purposefully created one meal plan for each gender category that contains dairy products to reflect two different (dairy vs. non-dairy) versions of the carnivore diet that exist. We also included liver as a food source in one of each of the gender category meal plans. We recognise that liver and other offal may not align with mainstream taste preferences; however, given their high micronutrient content and popularity amongst the carnivore population, we chose to include them, a decision reflecting a balance between optimal nutritional practices and the diverse reality of dietary preferences. Sample meal plans were entered into dietary analysis software Foodworks.online (Xyris Pty Ltd., Brisbane, Australia, 2024), which uses an Australian and New Zealand food database (NZ FOOD Files, 2018).

Meals were developed with a whole food principle (i.e., using foods that have been minimally processed) as a foundation. Mainly ruminant meats and their fats are suggested as providing the base of the carnivore diet, with limited quantities of pork, chicken, and fish. Versions of the diet also often feature eggs, offal, marrow, bone broth, and fermented dairy products. The diet plans aimed to reflect these different variations by including dairy and liver in separate meal plans.

## 3. Results

Table 3 presents the sample meal plan for the theoretical female and male case studies, each with a variation that contains dairy and one that contains liver. The nutritional analysis from these meal plans compared to the Australia/New Zealand NRVs is presented in Table 4.

NRVs (RDI or AI thresholds) were exceeded in both meal plans for riboflavin, niacin, phosphorus, zinc, Vitamin B6, Vitamin B12, selenium, and Vitamin A. The meal plans were below the RDI for the following micronutrients: thiamin, magnesium, calcium, Vitamin C, iron (meal plan 1 only), folate (meal plans 1 and 4 only), and iodine (meal plans 1 and 3 only). Potassium intake was below AI in meal plans 1, 2 (98% of AI), and 4. Fibre intake was significantly below AI in all meal plans (<1%).

The addition of dairy resulted in higher calcium intake for both the female and male meal plans (plan 1 and 3), albeit still below the RDI of 1000 mg/day (84 and 74% for the female and male, each respectively). The inclusion of liver in the female meal plan resulted in an iron intake which exceeded the RDI threshold. Meal plans which included iodised salt resulted in the RDI for iodine being exceeded, whereas without iodised salt, iodine intake only exceeded the RDI for the male meal plan (meal plan 3).

## 4. Discussion

The analysis shows that all four versions of the carnivore diet achieved most NRV thresholds. However, several nutrients fell short of the NRV recommendation and require careful consideration.

### 4.1. B Vitamins: Thiamin and Folate

Diets which exclude grain-based foods are frequently criticised for being inadequate in essential B-vitamins. Whist riboflavin, niacin, Vitamin B6, and Vitamin B12 were consistently sufficient across the meal plans, thiamin (Vitamin B1) was consistently insufficient. Thiamin is a water-soluble vitamin that is used in the body as a co-factor for cellular energy metabolism [24]. Inadequate thiamin intake can lead to the development of thiamin-deficiency diseases, including beri beri and Wernicke’s encephalopathy, both of which are considered rare diseases and only seen in extreme clinical situations such as alcoholism and severe malnutrition [21]. With inadequate thiamin, carbohydrate metabolism is impaired, resulting in the accumulation of pyruvate and lactate, which presents as intense vasodilation, elevated cardiac output and cardiac enlargement, and oedema [25].

A relationship between carbohydrate intake in the diet and thiamin requirements is well documented, but is conflicting and appears to depend on the type of carbohydrate. While diets containing whole grains also tend to contain adequate levels of thiamin, high doses of glucose have been shown to lead to acute thiamin deficiency in malnourished populations [24]. Furthermore, while rarely observed today, early reports of beri beri were originally described in populations consuming high quantities of polished grain [21]. It has been speculated that in the context of diets characterised by carbohydrate reduction, it may be that the requirement for thiamin is reduced due to a reduction in thiamin-requiring glycolytic metabolism [20]. However, given this theory has not been explored further in research, individuals following a carnivore diet and guiding nutritional professionals should be aware of the symptoms and potential risk of thiamin inadequacy. Although not shown in the case study meal plans, it is also possible for thiamin requirements to be sufficiently met by including pork; 280 g pork shoulder roast will provide 1.26 mg of thiamin (105% of the RDI for the male).

Without adequate planning/consideration, folate is another B-vitamin that could be inadequately consumed following a strict carnivore diet (as shown in two of the four meals plans). In its natural form, folate is predominantly sourced from green leafy vegetables and other plant foods. The synthetic form, folic acid, is also frequently fortified in cereal grains across Australia and New Zealand [21]. However, we showed in two of the meal plans, the folate requirements may be sufficiently met by including liver and eggs (i.e., 5 eggs provide 354.4 μg of folate, or 89% of the RDI for folate in the male case study). Therefore, individuals following the diet should look to include either of these foods to prevent folate inadequacy.

### 4.2. Vitamin C

Vitamin C is particularly interesting to discuss in the context of a carnivore diet. O’Hearn [14] presented empirical evidence and plausible mechanisms which could explain the apparent absence of Vitamin C deficiency diseases (scurvy) among groups of people consuming a largely animal based/carnivore diet. Despite being a poor source of Vitamin C, meat is recognised for having antiscorbutic (scurvy-preventing) properties [26]. It is suggested that scurvy may develop due to a lack of carnitine, which can be derived endogenously using Vitamin C, but is also available abundantly in meat. Thus, the large quantities of carnitine available in an animal-based diet may provide Vitamin C sparing effects [14]. However, research is still needed to confirm whether this theory is true, or whether some versions of the carnivore diet require Vitamin C supplementation.

### 4.3. Electrolytes: Calcium, Magnesium, Potassium, and Sodium

Calcium and magnesium intakes were below the RDI in all the theoretical meal plans; potassium intake was below the AI threshold in three of the four meals plan (one of which met 98%) and sodium exceeded the thresholds by a factor of 15–20 across the four plans. An increased risk for inadequate mineral intake (magnesium, potassium, calcium) while following a carnivore diet is frequently cited in community forums and lay websites, and therefore, supplementation with mineral electrolytes is common practice to address these inadequacies. However, due to there being multiple mechanisms through which calcium, magnesium, and potassium homeostasis is regulated (i.e., intestinal absorption, renal re-absorption/excretion, and bone resorption), acute deficiency diseases are uncommon [27].

Despite showing that calcium requirements could be met by including dairy products, there are some versions of the diet which exclude or limit dairy, and therefore calcium could be a nutrient of concern in these situations. Inadequate calcium intake may be a particular concern for bone health since a negative calcium balance can contribute to enhanced bone mineral losses [28]. Historically, low carbohydrate, high protein diets have been suggested to have a negative effect on calcium balance and, subsequently bone health due to an increased net acid load [29]. However, the evidence supporting the hypothesis that meat causes enhanced bone resorption has been heavily criticised [14,30]. Despite two meal plans providing inadequate calcium intake, it is plausible that several protective features of the carnivore diet could have an overall positive effect on bone health. First, protein intake itself also has a protective effect on bone and lean muscle mass; generally, higher intakes are positively associated with bone mineral density and protective against fractures [31,32]. Secondly, among individuals with type 2 diabetes, carbohydrate-reduced or ketogenic diets (including the carnivore diet) could have further protective effects for bone mineralisation by improving insulin sensitivity [33].

Subclinical magnesium deficiency is broadly recognised among the general population, and is particularly more common among people with type 2 diabetes and gastrointestinal diseases such as Crohn’s disease [27]. Inadequate magnesium intake can lead to weakness, loss of appetite, fatigue, nausea, and muscle spasms/cramps [27], the latter of which was identified as the most frequently reported adverse effect of the diet (4% of participants) in the survey by Lennerz, Mey [13]. Given that the carnivore diet may be considered specifically in the context of health conditions such as inflammatory bowel disease [34] and type 2 diabetes [13], these individuals are more likely to begin the diet with compromised nutrition status and could thus benefit from magnesium supplementation. Consistently inadequate magnesium intake remains a risk factor for magnesium deficiency. Thus, even healthy individuals following the diet for weight loss or aesthetic goals should consider supplementation.

Potassium and sodium homeostasis warrants discussion in the context of a carnivore diet. Potassium is typically found in large quantities in fruits and vegetables and therefore usually ingested with dietary nutrients that raise plasma glucose and stimulate insulin secretion. Insulin activates sodium-potassium-ATPase which pumps extracellular potassium into the tissues, especially muscle [35]. This insulin-stimulated potassium uptake has been shown to be preserved in individuals with type 2 diabetes, even when cellular glucose uptake is reduced [36]. A low glucose and insulin environment from following a ketogenic or carnivore diet could have potassium-sparing effects, thus potentially reducing dietary potassium requirements. Nonetheless, since this theory has not been tested in individuals following a strict carnivore diet, potassium supplementation is advisable.

The effect of insulin on sodium homeostasis is the reverse of potassium. Insulin inhibits renal sodium losses, thus in a low insulin environment, natriuresis is enhanced and dietary sodium needs may be increased [37]. For this reason, an increased sodium intake is generally advised on very low carbohydrate, ketogenic diet protocols, which includes the carnivore diet [38]. As such, the sample meal plans each include 3 tsp of salt across the day as a general starting point. Since sodium balance is influenced by a variety of factors, intakes should be adjusted based on taste, hydration status, and blood pressure (i.e., hypotension-related symptoms may suggest an increased sodium requirement) [38].

### 4.4. Iodine

For the last century, iodine fortification in table salt has been applied worldwide as a public health strategy for the prevention of iodine deficiency diseases [39]. New Zealand soils are recognised for being low in iodine, and therefore use of iodised salt is generally advised [21]. However, since a high salt intake is general advisable on very low carbohydrate, ketogenic diet protocols [38], using iodised salt, as we did in these meal plans, resulted in an exceptionally high iodine intake with 83% and 73% of the upper limit (UL) for iodine in meal plans 2 and 4. Both inadequate and excessive iodine intake can challenge thyroid function, which may be a particular concern for individuals with prior thyroid disorders [21]. Although the UL for iodine intake was not exceeded in the meal plans with three teaspoons of iodised table salt, individuals could mitigate a potential iodine excess by combining regular non-iodised with iodised table salt, for example, with a 2:1 ratio.

### 4.5. Iron

Iron intake was below the RDI in meal plan 1 (female). However, we also showed that female iron requirements could be met by including 80 g of lamb liver (meal plan 2). Whilst liver consumption is not necessarily appealing to the general population, it is more likely to be consumed in the context of the carnivore diet in the spirit of eating “nose to tail” for dietary variety.

The challenges associated with the RDI for iron have been discussed in the previous nutritional analysis of a carbohydrate-reduced diet [20]. In particular, iron bioavailability is affected by dietary composition and iron status. Plant-sourced phytates, polyphenols, tannins, and oxalates in the gut lumen reduce iron bioavailability [40]. Animal-sourced haem iron also has greater bioavailability than non-haem plant sources. It is plausible that should iron intake fall below the RDI (for females) in versions of the diet that exclude liver or mussels, consuming highly bioavailable iron sources without other competing antinutrients could sufficiently compensate for reduced net iron intake.

### 4.6. Fibre, “Anti-Nutrients”, and Bioavailability of Micronutrients

By completely excluding plants, the carnivore diet provides zero dietary fibre. There is no estimated average requirement (EAR) or RDI for dietary fibre; requirements are based on an AI (25 g/day for women and 30 g/day for men), which reflects the estimated average intake of apparently healthy people that is assumed to be adequate [21]. The key distinction between an AI and RDI is that with AI, the requirement is, essentially, assumed or optional. Thus, a central premise of the carnivore diet suggests that fibre is in fact nonessential, and that the removal of fibre and other plant compounds is beneficial [14].

One postulated benefit coming from the exclusion of fibre in the carnivore diet is a significant reduction in antinutrients. Antinutrients include lectins, oxalates, phytates, phytoestrogens, and tannins [41]. They are thought to restrict the bioavailability of various nutrients, and if absorbed in excess, may disrupt various physiological processes. For example, oxalate hyperabsorption, which occurs more commonly in individuals with gastrointestinal disorders, is associated with the formation of oxalate kidney stones [42]. Phytate, which is found in cereal grains, nuts, seeds, and legumes, acts as a strong chelator of mineral cations, impeding the bioavailability of copper, zinc, and iron in the diet [41]. Given that fibre and plant-sourced antinutrients compete with mineral absorption [43,44], it is plausible that not consuming plants could lead to lower net requirements of those nutrients.

From a gut and general health perspective, because of the lack of dietary fibre the carnivore diet is criticised for being unhealthy. Fibre is considered to be important for human health, due to the benefits of fibre on the gut microbial diversity and the products of microbial fermentation. Whilst change in dietary fibre intake can acutely alter microbiota composition, microbial diversity may be retained on an animal-based diet [45]. Short-chain fatty acids (SCFAs), which include acetate, propionate, and butyrate, are produced by microbial fermentation of indigestible carbohydrates (dietary fibre) [46]. Butyrate, in particular, serves as a primary source of energy for colonocytes, supporting the health and integrity of the intestinal lining, and has anti-inflammatory properties for the gut mucosa. Intestinal microbiota are also responsible for endogenous synthesis of a range of vitamins, including B vitamins and Vitamin K [47]. A recent analysis of genomes from microbiota found in the human gut suggested that changes in the gut microbiota composition have the potential to impact dietary B vitamin requirements [48]. Further research connecting dietary patterns to microbiome vitamin synthesis pathways is needed.

It has been postulated that in the context of a carnivore diet, the production of β-hydroxybutyrate (BHB), a ketone body produced by the liver might mitigate the lack of butyrate production from fibre fermentation [49]. BHB and butyrate, although chemically different molecules originating from distinct metabolic pathways, share several similar effects on the body, primarily due to their roles as signalling molecules and their influence on similar molecular pathways. BHB provides similar protective roles such as reducing inflammation and supporting cellular health. This overlap might explain why individuals on carnivore or ketogenic diets often report benefits such as reduced inflammation, improved mental clarity, and better metabolic health despite low fibre intake [13,50]. Based on our analysis, individuals following a carnivore diet may consider supplemental fibre intake to achieve the AI, however, dietary fibre may not be necessary in all cases and, theoretically, could off-set some of the intended effects of the diet itself.

### 4.7. Guidance for Nutrition Professionals

The carnivore diet is a contemporary nutrition strategy which may offer additional benefits beyond a diet that is just restricted in carbohydrate to individuals in unique clinical situations. However, given the current gap in evidence determining the impact of the diet on clinical nutrition parameters (i.e., serum markers), careful monitoring of nutrition status is advisable. Individuals following a strict carnivore diet for extended durations should be advised on how to identify clinical signs/symptoms of nutrient insufficiencies.

There are ways that intakes of several at-risk micronutrients could be optimised in the diet, as shown with the list of snacks and their corresponding nutrients in Table 5. However, it is important to note that versions of the diet which exclude or limit consumption some foods categories (i.e., dairy, pork, fish, and/or eggs) may have a higher risk of dietary inadequacies. These contexts, including offal and/or nutritional supplements may be necessary parts of the diet. Particular micronutrients of concern identified in our analysis include thiamin, folate, Vitamin C, calcium, magnesium, and potassium. Micronutrient supplementation is advisable, initially, during the first phases of the diet for those with prior health conditions [13], such as inflammatory bowel disease [34], that may mean poor nutrient status to start. Long-term use of the diet may also require additional supplementation, particularly when a limited variety of animal-based products is consumed.

## 5. Conclusions

The main limitation of this paper and the interpretation of the outcomes is the narrow existing literature base discussing the contemporary carnivore diet. Since there are multiple versions of a carnivore diet, and different options for potential food reintroduction i.e., some individuals return to varying amounts of plant inclusion once their inflammatory condition improves, our analysis does not necessarily capture the full picture of this dietary application. It is also important to recognise that single day analysis does not necessarily reflect average intakes across an entire week or month.

We have demonstrated various nutrient shortfalls of an entirely animal-based, carnivore diet as compared to the national NRV guidelines. Whilst it is hypothesised that metabolic changes associated with ketogenesis and the complete avoidance of plant-based foods could alter requirements for certain micronutrients, robust evidence is needed to confirm such speculation.

Finally, the metabolic health implications of a carnivore diet remain a topic of significant debate. While this diet may provide health improvements to those that require them, concerns about potential increases in lipid markers such as LDL cholesterol must be viewed in light of recent challenges to the diet-heart hypothesis and lipid hypothesis [51,52]. These long-standing theories, which link saturated fat and LDL cholesterol to cardiovascular disease, have been increasingly questioned, with emerging evidence suggesting a more complex relationship that may not support the traditional view of LDL cholesterol as a definitive risk factor for heart disease.

The carnivore diet might have its place in a small percentage of the population that has severe, unresolved inflammatory conditions. Rather than caution against such an eating approach, should it be helpful beyond other more acceptable restrictive dietary approaches, we stress that the nutrition gaps as shown in our findings require careful consideration, monitoring, and potential supplementation to avoid any long-term clinical nutrition deficiencies.

## Figures and Tables

**Table 1 nutrients-17-00140-t001:** Demographic characteristics for the theoretical case studies.

	Female	Male
Age (years)	19–50	19–50
Weight (kg)	59	69
Height (cm)	162	175
BMI (kg/m^2^)	22.5	22.5

**Table 2 nutrients-17-00140-t002:** Dietary energy and macronutrient parameters for the theoretical case studies.

	Female	Male
EER (kcal)	2120 (8870 kJ)	2663 (2663 kJ)
Fat (g, 70–75%E)	164.8–176.7	207.1–221.9
Protein (g, 25–30%E)	132.5–159.0	166.4–199.7
Carbohydrate (g, <5% E)	<26.5	<33.3

**Table 3 nutrients-17-00140-t003:** Sample meal plans.

Female		Male	
Meal Plan 1	Meal Plan 2	Meal Plan 3	Meal Plan 4
Breakfast Fried eggs (3, size 7 eggs) with leftover sauteed beef mince (90 g), cooked in butter (1 tbsp)	Breakfast Scrambled eggs (5, size 7 eggs) cooked in butter (2 tbsp), 2 rashers bacon (40 g)	Breakfast Scrambled eggs (5, size 7 eggs), cooked in 2 tbsp butter. Served with leftover grilled salmon (100 g)	Breakfast Leftover grilled scotch fillet (280 g), cooked in beef fat (2.5 tbsp)
Lunch Leftover beef mince patties (~150 g) with cheese slices (50 g)	Lunch Leftover lamb mince patties, containing 30% lamb liver (195 g), cooked with 2 tbsp lamb fat	Lunch Leftover beef mince patties (180 g), double cream brie cheese wedge (40 g)	Lunch Leftover chicken leg (1 thigh, 1 drumstick)
Dinner Grilled salmon fillet (210 g), cooked in own fat	Dinner Beef sirloin steak with visible fat (188 g cooked weight) cooked in 1.5 tbsp fat	Dinner Braised lamb neck chops (250 g), cooked in broth	Dinner Pork crackling/grilled pork fat trimmings (60 g) Bone broth with 1.5 tbsp beef fat
Snack Bone broth (1 cup) Unsweetened plain yoghurt (1 cup) Non-iodised salt, 3 tsp in meals across the day	Snack Bone broth (1 cup) Iodised salt, 3 tsp in meals across the day	Snack Bone broth (1 cup) Unsweetened plain yoghurt (1 cup) Non-iodised salt, 3 tsp in meals across the day	Snack Roast lamb leg (200 g), lambs fry (100 g) cooked in lamb/beef fat (3 tbsp)

Meal plan 1 and 3 include dairy. Meal plan 2 and 4 include offal and exclude dairy.

**Table 4 nutrients-17-00140-t004:** Nutrient analysis of carnivore meal plans.

	Female			Male		
Nutrient	Meal Plan 1	Meal Plan 2	NRV/Goal	Meal Plan 3	Meal Plan 4	NRV/Goal
Energy (kilocalories)	2180	2144	2120	2633	2662	2663
Fat (g) % TE	169.7 70.1	173.1 72.7	164.8–176.7 70–75	205.3 70.2	209.9 71.0	207.1–221.9 70–75
Protein (g) % TE	158.3 29.0	146.5 27.3	132.5–159.0 25–30	190.9 29.0	194.6 29.2	166.4–199.7 25–30
Carbohydrate (g) % TE	8.99 1.65	4.53 0.85	<26.5 <5	10.4 1.58	3.63 0.55	<33.3 <5
Saturated fat (g) % TE	61.8 25.5	75.6 31.7	<24 <10	86.3 29.5	88.6 29.9	<30 <10
Trans fat (g) % TE	3.01 1.24	5.40 2.27	<2.4 <1 _††_	5.41 1.85	5.91 2.00	<3 <1 *
MUFA (g)	63.2	59.3	NA	69.2	74.9	NA
PUFA (g)	17.1	9.79	NA	12.09	10.77	NA
Linoleic acid (Ʊ-6 PUFA) _†_	8.33	4.05	8	5.69	4.05	13
Alpha-linoleic acid (Ʊ-3 PUFA) _†_	2.11	1.17	0.8	1.70	1.34	1.3
Ʊ-3: Ʊ-6 ratio	3.95	3.46	10	3.35	3.02	10
Fibre (g) _†_	<1 *	<1 *	25	<1 *	<1 *	30
Thiamin (mg)	0.64 *	0.77 *	1.1	0.52 *	0.92 *	1.2
Riboflavin (mg)	2.47	3.92	1.1	2.91	4.90	1.3
Niacin (mg)	28.6	30.3	14	24.1	45.4	16
Vitamin C (mg)	1.22 *	16.8 *	45	1.21 *	33.2 *	45
Vitamin A (μg)	1046.6	26,320.2	700	1323.4	42,997.4	900
Vitamin E (mg) _†_	14.6	7.03	7	11.84	5.02 *	10
Vitamin B_12_ (μg)	14.3	51.7	2.4	15.45	80.6	2.4
Folate, total (μg)	244.7 *	472.2	400	409.7	33.42	400
Calcium (mg)	840.1 *	186.8 *	1000	764.4 *	76.2 *	1000
Iron (mg)	10.93 *	19.8	18	17.6	26.8	8
Magnesium (mg)	201.8 *	135.8 *	310–320	203.1 *	188.3 *	400–420
Zinc (mg)	19.8	18.2	8	28.2	30.3	14
Sodium (mg) _†_	7679.1	8789.4	460	8230.7	7200.2	460
Potassium (mg) _†_	2731.3 *	2006.5 *	2800	2702.9 *	2830.8 *	3800
Phosphorus (mg)	2027.0	1569.7	1000	1994.7	1910.9	1000
Selenium (μg)	99.0	99.5	60	111.4	96.1	70
Iodine (μg)	105.3 *	908.8	150	159.0	808.5	150

Grey shaded cells indicate NRV thresholds and macronutrient parameters (for energy, fat, carbohydrate, and protein) developed for the purposes of the theoretical carnivore diet case studies. * Nutrient reference value goal not met. _††_ WHO recommendations for trans fat. _†_ AI used where RDIs are unavailable. MUFA: Monounsaturated fatty acids; PUFA: Polyunsaturated fatty acids. NA: NRV thresholds not available for MUFA and PUFA.

**Table 5 nutrients-17-00140-t005:** Carnivore diet snacks and their key micronutrients.

Snack	Dominant Micronutrient(s)
Tinned fish with edible bones (i.e., sardines, salmon, mackerel)	Calcium, magnesium, long chain, omega 3 fatty acid (LCN3FA)
Smoked or pickled oysters or mussels	Iron
Boiled eggs	Folate, iron
Dried or smoked fish	LCN3FA
Liver pate	Iron, folate, Vitamin C
Bone broth, egg shell broth	Calcium, magnesium
Dairy products: Cultured yoghurt or kefir from cow’s milk Cheese	Calcium
Bone marrow	Vitamin A
Pork cold cuts (i.e., pork belly)	Thiamin

## Data Availability

Data available on request.

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
