# Peer review of "Assessing the Nutrient Composition of a Carnivore Diet: A Case Study Model"

_nutrients, 2024, doi:10.3390/nu17010140_

Round 1
Reviewer 1 Report
Comments and Suggestions for Authors
The authors of this submission have done a nice job exploring the potential nutritional adequacy of the novel carnivore diet. This research is timely and provides significant information to health care practitioners and dietitians as more and more individuals contemplate the fad diet. I believe the authors have presented unbiased research in an area where bias is rampant.
I have a few general considerations about the methods of this study, namely how many meal-plans the authors created/analyzed and how they were created. I am not sure that two total days of meal planning provides enough data to determine nutritional adequacy. Dietary patterns change from day to day, and this research does not showcase the ebbs and flow of eating in a natural environment. Therefore, I ask that the authors clarify if they created the meal plans to maximize the potential for dietary quality, or if the purpose of this study was to analyze the dietary patterns most likely to be replicated in a free-living environment. If the purpose of this study was to maximize dietary quality, then it would make most sense to incorporate some of the suggestions from the discussion (ie, lines 194-197, 202-206) into the initial analysis. If the purpose of this study was to maximize effectiveness of the “mainstream” carnivore diet, then more meal plans should be created to showcase the variability of the diet over more than two days.
The following are additional suggestions I have for the authors to consider:
Line 59: The Lennerz source title suggests that the data included is self-reported. Does this research also include food logs of what participants actually ate? It is my hypothesis that many individuals that are on a "carnivore diet" are not actually consuming only animal products. Therefore, in this instance, they are on a high-protein diet, which is different than a carnivore diet. Is there another source that you could use to support your claims within the paragraph?
Line 105-107: The authors state they derived their “protocol”/parameters of the carnivore diet from “anecdotal evidence, practice / experience from dietitians familiar with the approach, and practical food considerations.” Please elaborate on this process.
Line 112-113: Please clarify what this means. Previously, the authors stated the carnivore diet is a diet void of any plant products. This statement contradicts the essence of a carnivore diet; wouldn’t this just be a high protein diet then? If this statement is true for the “free-living” population, then this manuscript needs to clarify that this research is based upon the stricter definition of a carnivore diet.
Table 4: Is there a way to color code the table or add information about the degree to which nutrients meet nutrient requirements? For example, if nutrient requirements are met at 90% by the dietary plan, then add a column or row that showcases the nutrient requirement was met at 90%.
Line 248: Please cite this information. I might suggest altering this statement; many people choose to go on the carnivore diet for other purposes i.e. weight loss, muscle gain, aesthetics...
Lin 319: Because of this, does Vitamin K or B-vitamins intake change? Can this influence the total requirements of these vitamins and thus influence deficiencies? Please address this question.
Lines 359-362: Are there any studies that discuss the primary purposes for individuals following the carnivore diet? If so, please add.
Author Response
Comment 1:
The authors of this submission have done a nice job exploring the potential nutritional adequacy of the novel carnivore diet. This research is timely and provides significant information to health care practitioners and dietitians as more and more individuals contemplate the fad diet. I believe the authors have presented unbiased research in an area where bias is rampant.
I have a few general considerations about the methods of this study, namely how many meal-plans the authors created/analysed and how they were created. I am not sure that two total days of meal planning provides enough data to determine nutritional adequacy. Dietary patterns change from day to day, and this research does not showcase the ebbs and flow of eating in a natural environment. Therefore, I ask that the authors clarify if they created the meal plans to maximise the potential for dietary quality, or if the purpose of this study was to analyse the dietary patterns most likely to be replicated in a free-living environment. If the purpose of this study was to maximise dietary quality, then it would make most sense to incorporate some of the suggestions from the discussion (ie, lines 194-197, 202-206) into the initial analysis. If the purpose of this study was to maximise effectiveness of the “mainstream” carnivore diet, then more meal plans should be created to showcase the variability of the diet over more than two days.
Response 1: Thank you for this comment, it was important for us to be as unbiased as possible, it is pleasing to see this is acknowledged.
The purpose of our analysis was to capture the nutrient intake of a “typical” carnivore eating pattern while acknowledging that individuals may follow the diet with varying restrictions. The four authors of this paper are dietitians with clinical experience working with individuals who have used the diet to varying degrees. Based on our clinical perspective, the four meal plans (as shown in Table 3) were developed to represent a typical dietary pattern. While a broader sample of meal plans could show better variety in the context of “normal” eating, the strict carnivore diet generally has limited variability. Therefore, we deduced that four meal plans would adequately capture “usual” intake of someone following a carnivore diet.
The second reason why only four meal plans were analysed is that we chose to model this work on our two previous papers examining nutrient intakes of a low carbohydrate diet in adults (https://pubmed.ncbi.nlm.nih.gov/29439004/) and children (published in Nutrients), both of which included four meal plans.(https://pmc.ncbi.nlm.nih.gov/articles/PMC9032080/).
The following are additional suggestions I have for the authors to consider:
Line 59: The Lennerz source title suggests that the data included is self-reported. Does this research also include food logs of what participants actually ate? It is my hypothesis that many individuals that are on a "carnivore diet" are not actually consuming only animal products. Therefore, in this instance, they are on a high-protein diet, which is different than a carnivore diet. Is there another source that you could use to support your claims within the paragraph?
Response: Lennerz et al. specifically set out to examine behavioural characteristics and self-reported health status of individuals consuming the Carnivore Diet, which was defined as one which “aims to eliminate most or all plant-based foods” (p 2). Diets were reported from food frequency questionnaire data (“current intake frequency of main food groups and relevant food items” (p 2). Authors show that among their participant group, the frequency of plant matter intake is minimal but with some variability; for example, 1% of respondents consumed fruit weekly. Unfortunately, the reasons for diet variability among the survey respondents are not discussed in the paper but could be related to a range of hedonic/social or clinical (experiment, reintroductions) reasons. We would use another source to support our claims, however such literature does not yet exist, therefore we have elected to keep the reference to Lennerz et al. in this instance.
To make the context of the paper clearer, we have added a line in the text to clarify that participants reported consuming minimal to no plant matter (lines 59-63).
Line 105-107: The authors state they derived their “protocol”/parameters of the carnivore diet from “anecdotal evidence, practice / experience from dietitians familiar with the approach, and practical food considerations.” Please elaborate on this process.
Response: The authors of this paper consisted of four dietitians who have all worked with patients on using elimination diets, including the carnivore diet, for various clinical contexts. The anecdotal evidence used in developing the meal plans refers to common dietary patterns and macronutrient compositions of the diets from the wider community of individuals and health professionals familiar with the diet. We were required to use a combination of our professional experience and anecdotal evidence due to the contemporary nature of the carnivore dietary approach; there is no universal definition for the diet or specific parameters for macronutrient proportions (line 109-110).
Line 112-113: Please clarify what this means. Previously, the authors stated the carnivore diet is a diet void of any plant products. This statement contradicts the essence of a carnivore diet; wouldn’t this just be a high protein diet then? If this statement is true for the “free-living” population, then this manuscript needs to clarify that this research is based upon the stricter definition of a carnivore diet.
Response: Referring to line 112-113 in the original manuscript “Carbohydrate intake was set at <5% of energy intake, given that the diet typically includes minimal carbohydrate-containing foods”.
All animal-based foods are not necessarily devoid of carbohydrate, as we have shown in the meal plans which provided between 3.6 and 9.0 g of carbohydrate (between 0.5-2% of total energy). Small quantities of carbohydrate may be sourced on a true carnivore diet through trace amounts of glycogen in animal tissues (especially liver), dairy products (particularly milk and yoghurt), and in some processed or cured meats that contain sugars or fillers. Albeit, it should be noted that we did not include processed or cured meats in the sample meal plans as these may not be considered strictly compliant with the diet due to these additives.
Table 4: Is there a way to color code the table or add information about the degree to which nutrients meet nutrient requirements? For example, if nutrient requirements are met at 90% by the dietary plan, then add a column or row that showcases the nutrient requirement was met at 90%.
Response: We have realised that there was an error in the table note. Now, all nutrients that did not meet the NRV goal are indicated with asterisks. A different symbol has been used for the note “WHO recommendations for trans fat”. We decided that a simple asterisk was a clean way of showing the potentially problematic nutrients alongside their corresponding NRV goal. (Line 162-126)
Line 248: Please cite this information. I might suggest altering this statement; many people choose to go on the carnivore diet for other purposes i.e. weight loss, muscle gain, aesthetics...
Response: This comment refers to the sentence, “Given that the carnivore diet is often used in the context of health conditions such as digestive concerns and type 2 diabetes, these individuals are more likely to begin the diet with compromised nutrition status and could thus benefit from magnesium supplementation”. We have added references to Lennerz (2021) and a new citation to Norwitz & Soto-May (2024), which refer to individuals using the diet for specific health concerns. We have also added a sentence to acknowledge that “healthy” individuals following the diet for other purposes should also consider supplementation (line 262 to 264).
Line 319: Because of this, does Vitamin K or B-vitamins intake change? Can this influence the total requirements of these vitamins and thus influence deficiencies? Please address this question.
Response: This is a really interesting line of thought. Given the limited literature on the contemporary carnivore diet, we’re limited by speculation. We have added some text and two references relating to microbiota vitamin synthesis to consider this point in the discussion (line 338-342).
Lines 359-362: Are there any studies that discuss the primary purposes for individuals following the carnivore diet? If so, please add.
Response: We have modified this paragraph to include the reference to the Lennerz paper, which showed 93% of their respondents (total: 2029) were motivated primarily by health reasons (specified). We have also added a recent case report from Norwitz & Soto-Mota (2024) which presents ten cases of individuals with inflammatory bowel disease who have achieved clinical improvement with the diet (line 372).
Reviewer 2 Report
Comments and Suggestions for Authors
Thank you for the opportunity to participate in the revision of the article entitled “Assessing the nutrient composition of a carnivore diet: a case study model”.
The aim of this study is to assess the micronutrient adequacy of four versions of the carnivore diet in relation to national NRV thresholds. This is a diet that has recently attracted a great deal of interest and is based exclusively on the consumption of animal products, eliminating all plant antigens and xenobiotic secondary metabolites. Furthermore, the main aspect of the diet is the absence of carbohydrates. The work is interesting and well described, but I would recommend further investigation of some aspects to improve the manuscript.
As the authors point out, the argument against the consumption of fruit and vegetables is that they contain "anti-nutrients" that hinder the body's ability to digest certain nutrients. However, the simultaneous elimination of plant foods, which are rich not only in soluble and insoluble fiber, but also in valuable micronutrients and bioactive compounds (e.g., carotenoids, polyphenols, isothiocyanates, glucosinolates, etc.), may be a risk factor for the development of metabolic diseases such as obesity, diabetes, hypertension, dyslipidemia, but also some forms of cancer. There is a high risk of vitamin and mineral deficiencies, such as vitamin A, C, K, E, which are mainly found in plant foods. In addition, our microbiota needs fiber to produce compounds that help keep us healthy. In this context, the question arises: does the carnivorous diet consider the amount of nutrients obtained from vegetables and fruits, which far exceed the anti-nutrients provided by these foods?
Furthermore, although the authors have formulated dietary plans that somehow manage to cover the lack of some of these nutrients, including animal foods (e.g., eggs, liver, and dairy products) characterized by a low presence of the latter, the risks of the carnivorous diet must be considered. In addition to the lack of certain nutrients (especially vitamin C), this diet emphasizes the consumption of animal products, which are often rich in saturated fats and cholesterol, thus increasing plasma levels of LDL cholesterol (the so-called bad cholesterol), with a consequent increase in the risk of developing cardiovascular events. I therefore suggest that the discussion of the applicability of the carnivore diet be broadened, for example by including a paragraph on the possible negative effects of this diet, both in the short term (e.g., nausea, bad breath, etc.) and in the long term. This could help the reader to better understand the risks of following this diet.
Author Response
Comment 1: Thank you for the opportunity to participate in the revision of the article entitled “Assessing the nutrient composition of a carnivore diet: a case study model”.
The aim of this study is to assess the micronutrient adequacy of four versions of the carnivore diet in relation to national NRV thresholds. This is a diet that has recently attracted a great deal of interest and is based exclusively on the consumption of animal products, eliminating all plant antigens and xenobiotic secondary metabolites. Furthermore, the main aspect of the diet is the absence of carbohydrates. The work is interesting and well described, but I would recommend further investigation of some aspects to improve the manuscript.
As the authors point out, the argument against the consumption of fruit and vegetables is that they contain "anti-nutrients" that hinder the body's ability to digest certain nutrients. However, the simultaneous elimination of plant foods, which are rich not only in soluble and insoluble fiber, but also in valuable micronutrients and bioactive compounds (e.g., carotenoids, polyphenols, isothiocyanates, glucosinolates, etc.), may be a risk factor for the development of metabolic diseases such as obesity, diabetes, hypertension, dyslipidemia, but also some forms of cancer. There is a high risk of vitamin and mineral deficiencies, such as vitamin A, C, K, E, which are mainly found in plant foods. In addition, our microbiota needs fiber to produce compounds that help keep us healthy. In this context, the question arises: does the carnivorous diet consider the amount of nutrients obtained from vegetables and fruits, which far exceed the anti-nutrients provided by these foods?
Furthermore, although the authors have formulated dietary plans that somehow manage to cover the lack of some of these nutrients, including animal foods (e.g., eggs, liver, and dairy products) characterised by a low presence of the latter, the risks of the carnivorous diet must be considered. In addition to the lack of certain nutrients (especially vitamin C), this diet emphasises the consumption of animal products, which are often rich in saturated fats and cholesterol, thus increasing plasma levels of LDL cholesterol (the so-called bad cholesterol), with a consequent increase in the risk of developing cardiovascular events. I therefore suggest that the discussion of the applicability of the carnivore diet be broadened, for example by including a paragraph on the possible negative effects of this diet, both in the short term (e.g., nausea, bad breath, etc.) and in the long term. This could help the reader to better understand the risks of following this diet.
Response 1: Thank you for your comments surrounding the broad and widely discussed nutritional concerns with a carnivore diet. We aimed to examine the diet from an unbiased, neutral perspective to demonstrate nutrient adequacies and shortfalls of the diet; we were looking specifically at intakes of micronutrients relative to established nutrient reference values. A wider discussion of the role of plant bioactives in human health from both perspectives is not within the scope of this article. We also state that the theorised antigenic and xenobiotic properties of plants, particularly in the context of a strict carnivore diet, remain relatively unexplored in scientific literature.
Where you have mentioned high risk for vitamin and mineral deficiencies, we believe we have addressed these adequately in the analysis and discussion:
- Vitamin A: Requirements are comfortably met by the diet. Vitamin A sourced from animal products comes in the form of pre-formed Vitamin A, with greater bioavailability than pro-vitamin A carotenoids. We did not include this in the discussion since we chose to focus the discussion on nutrients which did not meet requirements.
- Vitamin C: We have discussed that Vitamin C falls short in the diet and the context around this.
- Vitamin K: We did not include Vitamin K in the analysis since it is generally accepted that Vitamin K is predominantly sourced from microbiota. The modified text (lines 338-342) refers to how potential changes in the microbiome through following a strict carnivore diet could alter requirements for postbiotics, including Vitamin K.
- Vitamin E: Vitamin E requirements are met in three of the four case studies.
- Fiber: We have discussed that fibre intake does not meet recommended levels in the diet and have also discussed how this low fibre intake may be required to achieve the intended effects of the diet.
Finally, regarding long-term health considerations and metabolic effects of the diet: Firstly and most importantly, we aimed to restrict the focus of this paper to nutrient status specifically. However, your suggestion to broaden the discussion on the potential negative effects of the carnivore diet is certainly valuable, as it would provide readers with a more comprehensive perspective. It is important to note that the claims regarding dietary fat, particularly saturated fats, cholesterol, LDL cholesterol, and their direct impact on cardiovascular events remain a topic of significant scientific debate. Recent research has challenged traditional views that high saturated fat intake and elevated LDL cholesterol levels directly increase cardiovascular risk. Emerging evidence suggests that the relationship is more nuanced, involving factors such as LDL particle size, overall metabolic health, and markers of inflammation. Furthermore, some studies have indicated that dietary patterns high in animal products but low in processed carbohydrates may not have the same cardiovascular implications as diets high in both saturated fats and refined carbohydrates.
Additionally, the carnivore diet's effects on health, including potential short-term issues like nausea and bad breath, and long-term outcomes, such as cardiovascular health or nutrient deficiencies, are not yet well-established in the scientific literature. Much of the existing evidence is anecdotal or based on small-scale studies, making it difficult to draw definitive conclusions.
Therefore, while including potential risks is important, it may also be beneficial to highlight the ongoing debates and gaps in the evidence regarding this diet. This balanced approach would enable readers to better understand the complexities and areas requiring further research.
As such, we have added the following sentences to the introduction and conclusion of the article.
Introduction, line 55-58: “It is also a common view that high consumption of animal products may have negative health effects due to changes in the microbiota (Huang, Cao et al. 2021), higher cardiovascular disease risk profile (Crimarco, Springfield et al. 2020), and other chronic health issues (Huang, Cao et al. 2021), however, the scientific basis for many of these potential negative effects is increasingly being questioned.”
Conclusion, line 389-396:
Finally, the metabolic health implications of a carnivore diet remain a topic of significant debate. While this diet may provide health improvements to those that require them, concerns about potential increases in lipid markers such as LDL cholesterol must be viewed in light of recent challenges to the diet-heart hypothesis and lipid hypothesis [52, 53]. These long-standing theories, which link saturated fat and LDL cholesterol to cardiovascular disease, have been increasingly questioned, with emerging evidence suggesting a more complex relationship that may not support the traditional view of LDL cholesterol as a definitive risk factor for heart disease.
Round 2
Reviewer 1 Report
Comments and Suggestions for Authors
The authors have made satisfactory edits to the manuscript where possible.